# A Randomized, Controlled Trial of Vitamin D Supplementation on Cardiovascular Risk Factors, Hormones, and Liver Markers in Women with Polycystic Ovary Syndrome

**DOI:** 10.3390/nu11010188

**Published:** 2019-01-17

**Authors:** Zeeshan Javed, Maria Papageorgiou, Harshal Deshmukh, Eric S. Kilpatrick, Vincent Mann, Lynsey Corless, George Abouda, Alan S. Rigby, Stephen L. Atkin, Thozhukat Sathyapalan

**Affiliations:** 1Department of Academic Diabetes, Endocrinology and Metabolism, Hull York Medical School, University of Hull, Hull HU3 2JZ, UK; Zeeshan.Javed@pkli.org.pk (Z.J.); m.papageorgiou@hull.ac.uk (M.P.); harshaldeshmukh@nhs.net (H.D.); 2Department of Pathophysiology and Allergy Research, Center of Pathophysiology, Infectiology, and Immunology, Medical University of Vienna, Vienna 1090, Austria; 3Department of Pathology, Sidra Medical and Research Centre, Doha PO Box 26999, Qatar; ekilpatrick@sidra.org; 4Gastroenterology Research Department, Hull Royal Infirmary, Hull HU3 2JZ, UK; Vincent.mann@hey.nhs.uk (V.M.); l.corless@hull.ac.uk (L.C.); George.Abouda@hey.nhs.uk (G.A.); 5Hull York Medical School, University of Hull, Hull HU3 2JZ, UK; A.rigby@hull.ac.uk; 6Weill Cornell Medical College Qatar, Education City, Doha PO Box 24144, Qatar; sla2002@qatar-med.cornell.edu

**Keywords:** polycystic ovary syndrome, vitamin D, liver markers, cardiovascular risk factors, hormones

## Abstract

Polycystic ovary syndrome (PCOS) increases the risk of metabolic syndrome and non-alcoholic-fatty-liver disease (NAFLD). Vitamin D supplementation may exert positive effects on liver biochemistry in patients with NAFLD; however, its effects on PCOS are unknown. This randomized, double-blind, placebo-controlled study explored the effect of vitamin D supplementation on cardiovascular risk factors (high-sensitivity C-reactive protein (*hs*-CRP), weight, body mass index (BMI), lipid profile, glucose levels, insulin levels, the homeostatic model assessment-insulin resistance (HOMA-IR), hormones (free androgen index (FAI), testosterone, sex hormone binding globulin (SHBG), and liver markers (alanine aminotransferase (ALT), hyaluronic acid (HA), N-terminal pro-peptide of type III procollagen (PIIINP), tissue inhibitor of metallo-proteinases-1 (TIMP-1), and the enhanced liver fibrosis (ELF) score). Forty women with PCOS were recruited and randomized to vitamin D (3200 IU) or placebo daily for 3 months. All outcomes were measured at baseline and 3 months follow-up (FU). Greater increases in vitamin D levels were shown in the supplementation group (vitamin D, baseline: 25.6 ± 11.4 nmol/L, FU: 90.4 ± 19.5 nmol/L vs. placebo, baseline: 30.9 ± 11.1 nmol/L, FU: 47.6 ± 20.5 nmol/L, *p* < 0.001). Between groups comparisons (% baseline change) revealed significant differences in ALT (*p* = 0.042) and a weak effect indicating a greater reduction in the HOMA-IR in the vitamin D group (*p* = 0.051). No further between group differences were seen in other cardiovascular risk factor, liver markers, or hormones. This study supports beneficial effects of vitamin D supplementation on liver markers and modest improvements in insulin sensitivity in vitamin D deficient women with PCOS.

## 1. Introduction

Polycystic ovary syndrome (PCOS), a common endocrine disorder, occurs in more than 10% of women of reproductive age and is associated with increased prevalence of cardiovascular risk factors (i.e., insulin resistance, hypertension, dyslipidemia), metabolic syndrome, and cardiovascular diseases [1,2]. As a hepatic manifestation of the metabolic syndrome, several studies suggest that women with PCOS are also at increased risk of developing non-alcoholic fatty liver disease (NAFLD) [3,4], which encompasses a spectrum of diseases progressing from simple steatosis to non-alcoholic steatohepatitis (NASH), and ultimately, cirrhosis [5]. In the general population, NAFLD appears to significantly exacerbate metabolic outcomes, and it is considered an independent risk factor for cardiovascular diseases [6,7]. Women with PCOS, and in particular those with insulin resistance, have the potential to develop a more severe and rapidly progressive liver disease at a young age, therefore, the early recognition of NAFLD and subsequent monitoring and management (especially given that NAFLD can be potentially reversed) in this population are considered of great clinical importance [5].

A recent meta-analysis demonstrated that vitamin D deficiency is common among women with PCOS, with 67–85% having serum concentrations of 25-hydroxyvitamin D (25OHD) <20 ng/mL (or 50 nmol/L) [8]. Low vitamin D correlates with obesity and it is associated with increased insulin resistance, testosterone, and dehydroepiandrosterone sulphate (DHEAS) levels [9]. Studies on vitamin D supplementation in PCOS have yielded mixed results, with some of them suggesting beneficial effects on glucose metabolism and insulin resistance [10], especially when vitamin D is given continuously at lower (<4000 IU/day) doses as suggested in a recent meta-analysis [11], and improvements in menstrual frequency [12,13] and hyperandrogenism [13,14,15], whereas others demonstrate no significant improvements [16,17,18,19,20]. Emerging evidence has associated NAFLD with vitamin D deficiency, and in epidemiological studies, both conditions share multiple cardiovascular risk factors [21,22]. Some vitamin D supplementation trials have shown positive effects on hepatic steatosis, liver-related biochemical markers, or both in patients with NAFLD [23,24,25]; however, its effects in women with PCOS have not been previously explored.

Although liver biopsy is the gold standard for NAFLD diagnosis and staging, as well for evaluating the efficacy of potential treatments, its use is limited by cost, sampling errors, and procedure-related morbidity and mortality [5]. Consequently, there is consensus that non-invasive markers should be the first choice to exclude significant fibrosis due to their good inter-laboratory reproducibility, high reliability, and widespread availability [26]. The enhanced liver fibrosis (ELF) test is one of the first commercially available serum multi-marker fibrosis tests [27]. It has been validated in various patient groups in a large multicenter study and has been found to be as accurate as liver biopsy at predicting liver disease-related outcomes [28,29,30].

The aim of this study was to explore and compare the effects of vitamin D supplementation vs. placebo on cardiovascular risk factors, hormones, and markers of liver injury and fibrosis in vitamin D overweight and obese vitamin D deficient women with PCOS.

## 2. Materials and Methods

This double-blind, randomized, placebo-controlled study was performed in the Academic Diabetes, Endocrinology, and Metabolism Unit, Hull University Teaching Hospitals NHS Trust, Hull, United Kingdom. Fifty-four reproductive-aged women (18–45 years) diagnosed with PCOS based on all three diagnostic criteria of the Rotterdam consensus were screened for vitamin D deficiency [31]. Non-classical 21-hydroxylase deficiency, hyperprolactinemia, Cushing’s disease, and androgen-secreting tumors were excluded by appropriate tests. The participants were considered vitamin D-deficient, when their serum 25OHD levels were less than 20 ng/mL (<50 nmol/L). We excluded PCOS women who: (1) had type 2 diabetes (all women had an oral glucose tolerance test) or thyroid disorders, (2) were on medication that could interfere with calciotrophic hormones for the preceding 6 months, (3) were planning to conceive or were using any oral or implantable contraceptives or any other treatments likely to affect ovarian function, insulin sensitivity, or lipids for at least 3 months before entering the study, but a stable dose of metformin for 3 months was allowed, (4) had nephrolithiasis or conditions resulting in hypercalcemia or hypercalciuria, and (5) had known hypersensitivity to vitamin D or any of the excipients in the product and/or peanut or soya allergy. None of the participants had NAFLD on ultrasound. All patients gave their written informed consent before partaking. The study was given ethical approval by the NRES Committee Yorkshire and the Humber - Leeds East. All methods were performed in accordance with the local Trust Research and Development guidelines and regulations. This study was registered in the ClinicalTrials.gov registry as NCT02513381.

Participants attended three visits (screening visit, baseline visit, 3 months follow-up). In the screening visit, participants were screened against inclusion and exclusion criteria by medical history, clinical examination, and routine blood tests. Forty eligible patients were randomly assigned to take vitamin D 3200 IU (Swiss Caps AG, Cornu, Switzerland) or placebo (Swiss Caps AG, Kirchberg, Switzerland) in capsule form once daily based on a computer-generated randomization list by Essential Nutrition Ltd., U.K., and attended a baseline visit (baseline, initiation of intervention) and a follow-up visit 3 months following vitamin D supplementation or placebo administration. The supplementation arm was based on the National Osteoporosis Society, U.K. guidelines which recommend treating patients who have serum 25OHD levels between 30–50 nmol/L and are at increased risk of developing vitamin D deficiency in the future because of reduced exposure to sunlight [32]. High-sensitivity C-reactive protein (hs-CRP), lipid profile, glucose levels, insulin levels, and the homeostatic model assessment-insulin resistance (HOMA-IR) were assessed as the primary outcome measures. As secondary outcomes, other cardiovascular risk factors (weight, body mass index (BMI)), hormones (free androgen index (FAI), testosterone, sex hormone binding globulin (SHBG)), and liver markers (alanine aminotransferase (ALT), hyaluronic acid (HA), N-terminal propeptide of type III procollagen (PIIINP), tissue inhibitor of metallo-proteinases-1 (TIMP-1), and the (ELF) score) were evaluated. All participants were advised to maintain their usual dietary and lifestyle habits during the study, including smoking, sunlight exposure, and physical activity that could affect the levels of vitamin D and metabolic indices. The compliance with treatment was calculated by counting the returned medications.

### 2.1. Study Measurements

During all visits, participants were weighed barefoot (Marsden Weighing Machine Group Ltd., Rotherham, UK). Blood pressure was measured using an automated device (NPB-3900; Nellcor Puritan Bennett, Pleasanton, CA, USA); for this measurement, subjects were seated quietly for at least 5 min and with the right arm supported at heart level. Three readings were taken, each at least 2 min apart, and then the mean value of the readings was calculated. Following an overnight fast, venous blood samples were collected and placed into fluoride oxalate and serum gel tubes. Samples were separated by centrifugation at 2000 g for 15 min at 4 °C, and the aliquots were stored at −80 °C within 1 h of collection. Overnight urine samples were collected and aliquots were stored at −80 °C until batch analysis. Serum insulin was assayed using chemiluminescent immunoassay on the Beckman Coulter UniCel DxI 800 analyzer (Beckman Coulter UK Ltd., High Wycombe, UK). Plasma glucose was measured using a Beckman AU 5800 analyzer (Beckman-Coulter, High Wycombe, UK) and according to the manufacturer’s recommended protocol. Insulin resistance was computed as HOMA-IR = Glucose × Insulin/22.5 [33]. Serum vitamin D levels and testosterone were quantified using isotope-dilution liquid chromatography tandem mass spectrometry (LC-MS/MS). SHBG was measured using a chemiluminescent immunoassay on the UniCel DxI 800 analyzer (Beckman-Coulter, High Wycombe, UK), applying the manufacturer’s recommended protocol. The FAI was calculated as: (total testosterone/SHBG) x 100. Total cholesterol, triglycerides, high-density lipoprotein cholesterol (HDL-C), ALT, and *hs*-CRP levels were measured enzymatically using a Beckman AU 5800 analyzer (Beckman-Coulter, High Wycombe, UK). Low-density lipoprotein cholesterol (LDL-C) was calculated using the Friedewald equation. The ELF score provides a single value using an algorithm combining serum measurements of three biomarkers including HA, TIMP-1, and PIIINP. We used the ADVIA Centaur XP/XPT immunochemical analyzer (Siemens Healthcare Diagnostics Inc., Erlangen, Germany) according to manufacturer’s instructions. The ELF score was calculated directly by the instrument employing the following equation [30]:ELF score = 2.278 + 0.851 ln(C_HA_) + 0.751 ln(C_PIIINP_) + 0.394 ln(C_TIMP-1_)

### 2.2. Statistical Analysis

Based on our previous work [34], this study was powered to detect differences in *hs*-CRP. The minimum difference worth detecting/observed difference was 32.7%, with an effect size of 1.11; therefore, for 90% power and a significance level of 5%, a sample size of 16 per group was calculated. Adjusting for a possible 20% dropout rate meant a total of 40 patients need to be recruited (nQuery, Statistical Solutions, Saugus, MA, USA). To negate the differences in the baseline values of the two groups, data were expressed as percentage change from baseline for all variables. The effects of treatment were evaluated by detecting differences in percentage changes between groups using an independent t-test or the Mann-Whitney test. Data comparisons between baseline and follow-up at 12 weeks within groups were carried out using the paired t-test (normally distributed data) or the Wilcoxon signed rank test for non-normally distributed data. Data with normal distribution are presented as means ± SD, while data following a skewed distribution are presented as medians with interquartile range. All statistical analyses were performed using SPSS version 25.0 (IBM SPSS Statistics, Chicago, IL, USA).

## 3. Results

Thirty-seven patients (vitamin D group, *n* = 18; placebo group, *n* = 19) completed the 3-month study period. Two patients from the vitamin D group and one patient from placebo group dropped out of the study (Figure 1). After their exclusion, compliance was 99% in both groups by counting returned medication. None of the subjects developed any significant side effects over the course of the study. The mean age group of patients was 28.6 ± 6.4 years (vitamin D 28.6 ± 5.5 vs. placebo 29.1 ± 7.5 years). The baseline vitamin D levels, anthropometric, hormonal, and biochemical parameters of the two groups are given in Table 1.

The study started in July 2015 and the last patient’s last visit was in November 2016. Vitamin D levels significantly increased at 3 months compared to baseline in both groups, with greater increases shown in the women with PCOS who were randomized to the vitamin D supplementation (vitamin D group, baseline: 25.6 ± 11.4 nmol/L, 3-month follow-up: 90.4 ± 19.5 nmol/L vs. placebo group, baseline: 30.9 ± 11.1 nmol/L, 3-month follow-up: 47.6 ± 20.5 nmol/L, *p* < 0.001), confirming adherence to supplementation.

There were no significant changes in weight, BMI, blood pressure, hs-CRP, lipid profile (total cholesterol, LDL-C, HDL-C, TG), markers of insulin sensitivity (plasma glucose levels, insulin levels, HOMA-IR), FAI, testosterone, or SHBG within vitamin D supplementation or placebo group. Expressed as percentage change from baseline, there was a week effect, indicating a greater reduction in HOMA-IR in the vitamin D group (*p* = 0.051), but no significance between group differences were shown for any of these variables (Table 1).

Following 3 months of supplementation, the vitamin D group experienced significant reductions in ALT levels (*p* = 0.042), HA levels (*p* = 0.019), and the cumulative ELF score (*p* = 0.022). Within the placebo group, ALT levels increased at 3 months follow-up (*p* = 0.039), with no further changes in any liver marker (Table 1). Between group comparisons revealed a difference in ALT (% baseline change) (*p* = 0.001). Mean values of percentage change from baseline for HA, PIIINP, and the ELF score were reduced in the vitamin D group and increased in the placebo group after 3 months of supplementation; however, between groups comparison did not reach statistical significance (Table 1).

Given that some participants in the vitamin D group (*n* = 3) and the placebo group ( *n* = 8) were on metformin, which may interfere with our results, we repeated between group comparisons after excluding data from participants taking metformin, and the results remained unchanged (Appendix A). 

## 4. Discussion

In this study, we demonstrate that compared to placebo, vitamin D supplementation resulted in modest improvements in ALT and insulin resistance, whereas no further between-group differences were seen in cardiovascular risk factors or hormones. Within group comparisons also showed that supplementation with vitamin D (3200 IU/day) over a 3-month period resulted in significant improvements in individual liver markers (ALT, HA,) and in the ELF score compared to baseline in overweight and obese vitamin D deficient women with PCOS.

There is evidence to suggest that women with PCOS are more susceptible to NAFLD than BMI-matched controls [3,4], and despite the reversibility of the disease, treatment options are largely unexplored in this population. In the present vitamin D supplementation study, we assessed liver fibrosis by HA, PIIINP, and TIMP-1, each of which reflect ongoing sinusoidal fibrogenesis and fibrolysis in the liver, and their composite ELF score [21]. The ELF score can accurately differentiate mild (<7.7), moderate (≥7.7 to <9.8), and severe fibrosis (≥ 9.8), and is suggested to be a prognostic tool with good predictability of clinical outcomes [27,28]. In our study, the treatment group had a reduction in ALT levels and went from a moderate to a mild fibrosis category, as indicated by a significant drop in the ELF score post-intervention (within group comparison). In contrast, ALT levels increased in the placebo group within the timeframe of the study. The reason behind this increase in the placebo group is unclear, and may involve a worsening metabolic profile due to the PCOS progression. Whereas there are no comparative data from vitamin D supplementation studies on liver markers in PCOS, the positive changes in markers of liver fibrosis in the present study are in accordance with relevant animal studies demonstrating beneficial effects of vitamin D on liver histology [35,36], but also human studies in patients with NAFLD [23,24,25,37]. For example, Papapostoli et al. demonstrated favorable alterations in transient elastography (controlled attenuation parameter) after a 6-month vitamin D treatment (20,000 IU per week) [23]. By using liver biopsy, Geier et al. also suggested a trend towards reduced hepatic steatosis in patients with histologically determined NASH, treated with daily vitamin D (2100 IU) for 48 weeks, although due to the limited number of specimens, these results did not reach significance [25]. In agreement with our ALT results, in the same study, patients in the vitamin D group had significantly reduced ALT levels after the end of the supplementation period compared to placebo [25]. Similarly, Dasarathy et al., reported improvements in ALT levels only in responders to vitamin D supplementation, defined as those whose vitamin D levels were >30 ng/ml following supplementation with 2000IU vitamin D daily for 6 months [38]. Two other trials also showed greater reductions in ALT levels in NAFLD patients who were randomized to a lifestyle modification program plus vitamin D supplementation, compared to those who received the lifestyle modification programs only [24,37].

Several potential mechanisms have been suggested by which vitamin D reduces the hepatic steatosis and fibrosis in NAFLD patients [17,33]. The key process that leads to the development of liver fibrosis in NAFLD patients is the activation of hepatic stellate cells, by fatty infiltration, with subsequent excessive deposition of extracellular matrix and destruction of normal liver architecture [21]. Recent studies have shown that hepatic stellate cells express vitamin D receptors, which when activated, can inhibit hepatic stellate cells proliferation by suppressing fibrotic gene expression [39]. These data are in accord with in-vitro studies on human cells [40] and animal models, showing that vitamin D treatment inhibited fibrogenesis [41] and suppressed hepatic steatosis and fibrosis by enhancing hepatic autophagy [42]. Other proposed mechanisms that also link vitamin D with NAFLD include improvements in insulin sensitivity and reduction of inflammation [21,22]. Indeed, some clinical trials have shown that treatment with vitamin D improves parameters of insulin sensitivity in patients with NAFLD [24], a finding consistent with the weak effect indicative of a reduction in the HOMA-IR shown in the present study. At a molecular level, these effects may be due to vitamin D-related enhancements in glucose transport in muscles, upregulation of the glucose transporter type 4 (GLUT4) expression and translocation in cell surfaces, with subsequent glucose use by adipocytes [22]. Reduction of inflammation may also explain the effects of vitamin D on improving NAFLD. For instance, artificial lighting in rats decreased the degree of inflammation and hepatic apoptosis. In humans, some studies suggest reductions in markers of inflammation and oxidative stress following vitamin D supplementation [43], although we, as well others [44,45], did not show any vitamin D effects on *hs*-CRP.

With the exception of a weak effect indicating a reduction in HOMA-IR in the supplementation group compared to the placebo group, vitamin D supplementation had no significant effect on cardiovascular risk factors, including body weight, glucose and insulin levels, and lipid profile. Vitamin D supplementation studies have reported positive [10,11,12,13,14] or no changes in metabolic markers [16,17,18,19,20] in this population. In accordance with the HOMA-IR results in our study, a recent meta-analysis of RCTs suggested that the HOMA-IR also declined significantly when vitamin D was supplemented at doses lower than 4000 IU/day [11]. Conversely, Asemi et al. explored the effects of 50,000 IU vitamin D per week over 8 weeks in PCOS and found no significant differences in insulin resistance or blood lipids compared to placebo [20]. Similarly, Garg et al. revealed no differences in measures of insulin resistance and sensitivity in PCOS women randomized to either 120,000 IU of cholecalciferol monthly or placebo over 6 months (PCOS women in both groups also received 1500 mg of metformin daily) [18]. We also showed no significant alterations in FAI, testosterone, or SHBG in PCOS in response to vitamin D supplementation; these findings are consistent with two recent meta-analyses on the vitamin D effects in PCOS, which revealed no change in testosterone or SHBG levels [46,47]. Contrary to these findings, reductions in FAI and testosterone and elevations in SHBG were seen in women with PCOS following daily supplementation with a “high vitamin D dose” (4000 IU/day), but not a “lower dose” (1000 IU/day), over a 12-week period in a RCT that was not included in these meta-analyses [15]. These findings underscore the current lack of consensus on the effects of vitamin D in PCOS, which can be largely explained by the heterogeneity in study designs, sample sizes, characteristics of vitamin D supplementation (i.e., monotherapy or add-on therapy, duration, frequency and dosage), and baseline vitamin D status.

Our study is strengthened by our design (double blind, randomized, placebo-controlled trial), the low dropout rate, and the high level of compliance with supplementation (99%). In addition, to detect early liver fibrosis in our population we relied on the assessment of multiple markers of liver fibrosis in blood samples and the subsequent estimation of the ELF score; this method is less onerous than a liver biopsy, but evidence-based in characterizing NAFLD severity and predicting NAFLD-related outcomes [28,29]. We should also acknowledge our study limitations. The inclusion of an additional approach to quantify liver fibrosis (i.e., the “gold standard” liver biopsy or other non-invasive method) would be useful in confirming our findings and would provide therapeutic insights into liver injury in this understudied health aspect in PCOS. Our study population was relatively healthy, without severe liver fibrosis or metabolic abnormalities; thus, our findings may be less relevant for women with PCOS with progressed NAFLD or type 2 diabetes. Vitamin D levels were raised in the supplementation group, but also, albeit to a less degree, in the placebo group, due to seasonal variations. Nevertheless, at 3 months follow-up, mean vitamin D levels in the supplementation group were well within the normal range, but mean vitamin D levels in the placebo group remained within the insufficiency range. A stable dose of metformin, which has been shown to improve biochemical and metabolic variables in NAFLD [48], was an inclusion criterion in our study. It is, however, unlikely that metformin contributed to the reported vitamin D effects, since there were more patients on stable doses of metformin in the placebo group (*n* = 8), rather than in the vitamin D group (*n* = 3). The notion that metformin did not interfere with our results is further supported by the absence of changes in the results of the between group comparison after excluding the participants taking metformin in both groups.

## 5. Conclusions

In conclusion, this study supports beneficial effects of vitamin D supplementation on markers of liver injury and fibrosis, and modest improvements in insulin resistance in overweight and obese vitamin D deficient women with PCOS. In contrast, other cardiovascular risk factors and hormones did not change in response to the 3-month vitamin D supplementation. These results need confirmation in future larger-scale trials with a longer supplementation period conducted in women with PCOS at different NAFLD stages.

## Figures and Tables

**Figure 1 nutrients-11-00188-f001:**
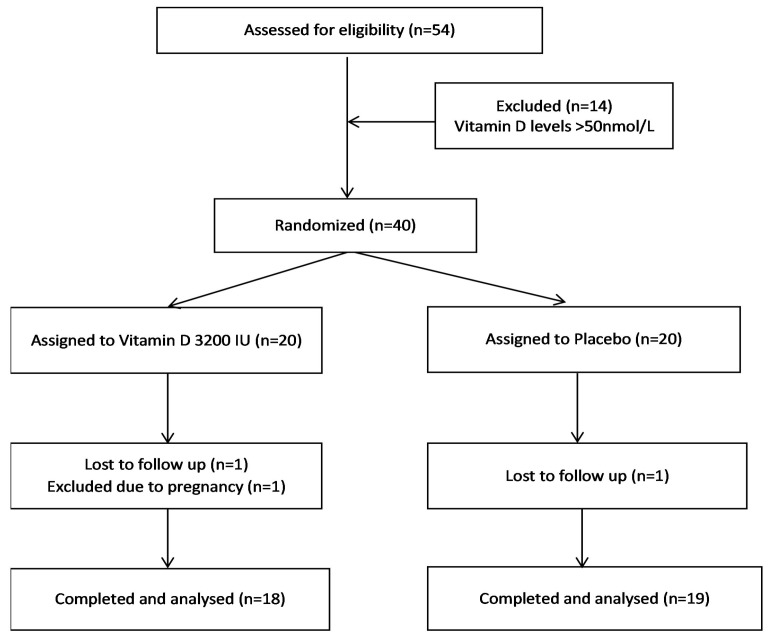
Flow chart showing the progress of patients throughout the trial.

**Table 1 nutrients-11-00188-t001:** Comparison of vitamin D levels, cardiovascular risk factors, hormones, and liver markers after vitamin D or placebo supplementation.

	Vitamin D group (*n* = 18)			Placebo group (*n* = 19)			% Change		
Parameter	Baseline	3 months	*p*-value ^a^	Baseline	3 months	*p*-value ^a^	Vitamin D	Placebo	*p*-value ^b^
25OHD (nmol/L)	25.6±11.4	90.4 ± 19.5	**<0.001***	30.9 ± 11.1	47.6 ± 20.5	**<0.001 ***	319 ± 214	59.5 ± 56.7	**<0.001 ****
Weight (kg)	97.9 ± 31.3	98.2 ± 31.9	0.75	93.8 ± 22.3	94.3 ± 21.5	0.51	1.6(4.1)	0.7(3.7)	0.87
BMI (kg/m^2^)	35.4 ± 10.6	35.5 ± 10.8	0.76	33.8 ± 7.2	34.0 ± 7.0	0.44	1.7(4.1)	0.7(3.7)	0.92
SBP (mmHg)	117 ± 9.0	116 ± 14.6	0.68	121 ± 15.6	122 ± 12.5	0.75	−0.9 ± 7.8	1.1 ± 7.6	0.42
DBP (mmHg)	69.0 (12.5)	73.0 (18.8)	0.55	80.9 ± 10.7	80.6 ± 10.7	0.1	1.7 ± 11.8	-0.2 ± 6.9	0.55
hs-CRP (mg/L)	2.6 (7.9)	2.3 (6.3)	0.69	2.7 (8.1)	3.4 (8.5)	0.14	17.7 ± 67.0	15.6 ± 45.5	0.91
TC (mmol/L)	4.9 (1.0)	4.9 (1.1)	0.24	4.9 ± 0.8	4.9 ± 0.8	1.00	−2.2(19.9)	5.3(14.9)	0.22
LDL-C (mmol/L)	3.0 ± 0.7	3.1 ± 0.7	0.29	3.0 ± 0.7	2.9 ± 0.6	0.65	7.3(19.3)	−3.3(26.9)	0.26
HDL-C (mmol/L)	1.4 ± 0.3	1.3 ± 0.4	0.88	1.2 (0.5)	1.1 (0.4)	0.11	0.0(17.4)	0.0(6.3)	1.0
TG (mmol/L)	1.2 (0.8)	1.1 (0.9)	0.46	1.1 (0.8)	1.1(1.0)	0.12	13.8(45.3)	5.9(71.3)	0.39
Fasting glucose (mmol/L)	4.7(0.5)	4.6(0.7)	0.30	4.8 ± 0.4	4.8 ± 0.5	0.76	3.0 ± 8.9	0.7 ± 7.6	0.55
Fasting insulin (µIU/mL)	14.2 (12.8)	12.3 (17.1)	0.50	11.7 ± 6.5	12.8 ± 8.0	0.39	11.2 ± 43.2	16.3 ± 45.0	0.73
HOMA-IR	2.9 (2.8)	2.5 (3.9)	0.44	2.1 (2.1)	2.2 (2.8)	0.31	−16.3 ± 52.5	19.3 ± 54.3	0.051
FAI	5.3 (5.8)	4.9 (5.5)	0.31	5.5 ± 3.1	5.9 ± 3.4	0.42	−3.1(48.3)	6.3(41.6)	0.26
Testosterone (nmol/L)	1.1 (0.9)	1.1 (0.8)	0.27	1.1 ± 0.4	1.2 ± 0.4	0.18	−0.6 ± 45.3	11.3 ± 27.3	0.34
SHBG (nmol/L)	24.0 (24.8)	24.5 (19.8)	0.72	19.0 (17.0)	20.0 (13.0)	0.57	1.0 ± 16.1	2.8 ± 17.9	0.74
ALT (IU/L)	27.0 ± 12.2	22.1 ± 11.5	**0.042 ***	24.9 ± 14.9	28.2 ± 14.4	**0.039 ***	−16.7 ± 25.7	18.6 ± 28.6	**0.001 ****
HA (ng/mL)	18.5 ± 9.9	12.6 ± 6.7	**0.019 ***	12.4 (9.4)	12.9 (12.7)	0.84	−18.7 ± 48.0	9.4 ± 58.6	0.12
PIIINP (ng/mL)	7.5 ± 2.0	6.5 ± 2.0	0.78	6.6 (2.5)	6.5 (2.2)	0.66	−10.2 ± 31.2	4.6 ± 30.3	0.15
TIMP-1 (ng/mL)	156 (55.4)	154 (55.3)	0.45	164 ± 42	154 ± 44	0.19	−0.9(54.1)	−8.3(24.3)	0.71
ELF Score	8.1 ± 0.5	7.6 ± 0.8	**0.022 ***	7.8 ± 0.6	7.7 ± 0.7	0.68	−5.9 ± 10.2	−0.6 ± 9.7	0.11

Data are presented as mean ± SD if normally distributed, or median (interquartile range) if not normally distributed. **^a^** indicates *p*-values for within groups comparisons; **^b^** indicates *p*-values for between group comparisons. Bold data indicate statistical significant p-values; * *p* < 0.05, significant different compared to baseline within group, ** *p* < 0.05, significant difference between groups. 25OHD: 25-hydroxyvitamin D; BMI: body mass index; SBP: systolic blood pressure; DBP: diastolic blood pressure; *hs*-CRP: high sensitivity-C-reactive protein; TC: total cholesterol; LDL-C: low density lipoprotein cholesterol; HDL: high density lipoprotein cholesterol; TG: triglycerides; FAI: free androgen index; SHBG: sex hormone binding globulin; HOMA-IR: homeostatic model assessment of insulin resistance; ALT: alanine aminotransferase; HA: hyaluronic acid; PIIINP: amino-terminal propeptide of type III procollagen; TIMP-1: tissue inhibitor of metallo-proteinases-1; ELF Score: enhanced liver fibrosis.

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
