# Peer review of "A Randomized, Controlled Trial of Vitamin D Supplementation on Cardiovascular Risk Factors, Hormones, and Liver Markers in Women with Polycystic Ovary Syndrome"

_nutrients, 2019, doi:10.3390/nu11010188_

Round 1
Reviewer 1 Report
What is the reason behind the significant short term increase in the ALT in placebo group. Does the placebo drug have an effect on the ALT levels? This significant increase have not been properly discussed.
Please find attached and make the corrections accordingly.

Author Response
We would like to thank both reviewers for their useful comments on our paper, which substantially improved our manuscript. We have made several changes to the manuscript as a result, with the changes highlighted in yellow in the manuscript file for ease of tracking. A point-by-point response to the reviewer comments is provided below.
Reviewer 1
What is the reason behind the significant short term increase in the ALT in placebo group. Does the placebo drug have an effect on the ALT levels? This significant increase have not been properly discussed.
Thank you for your comment. Given that women with PCOS in the placebo group received a placebo capsule based on maize, which did not contain any vitamin D and thus, it does not affect liver function, we speculate that PCOS per se resulted in modest increases in ALT levels within the timeframe of the present study. Future longitudinal studies, where PCOS related outcomes such as biochemical markers are assessed at frequent intervals are needed to give further insights into how ALT levels change with the progression of PCOS.
We have added a brief discussion to address this point in the manuscript:
Page 7, lines 215-218 (revised manuscript): “In contrast, ALT levels increased in the placebo group within the timeframe of the study. The reason behind this increase in the placebo group is unclear, but may involve a worsening metabolic profile due to the PCOS progression per se”.
Page 2, line 79: replace using with administering- Page 2, line 81-83 (revised document)-amended in the text to reflect the setting and location of this RCT- “This double-blind, randomized, placebo-controlled study was performed in the Academic Diabetes, Endocrinology and Metabolism Unit, Hull University Teaching Hospitals NHS Trust, Hull, United Kingdom”.
Page 2, line 91: You excluded the patients with diabetes and allows the ones with metformin (are they using metformin for any other reason). The metformin administered can interfere the results.
All participants who were on metformin in the present study had Hba1c levels less than 42 mmol/mol. We have acknowledged that this is a limitation of the current study (page 8, line 278-282). Additionally, we have rerun the between group comparisons after excluding the participants on metformin in both groups and our results did not change, suggesting that metformin did not interfere with our intervention in the present study.
We have added relevant descriptions in our results and discussion:
Page 5, lines 191-194 (revised manuscript): “Given that some participants in the vitamin D (n=3) and the placebo group (n=8) were on metformin, which may interfere with our results, we repeated between group comparisons after excluding data from participants taking metformin and the results remained unchanged (data not shown)”.
Page 8, lines 294-296 (revised manuscript): “The results of the between group comparison remained unchanged after excluding the participants taking metformin in both groups, further supporting the notion that metformin did not interfere with our results”.
Page 3, line 107: replace with including Page 3, line 117 (revised document)-amended in the text
Page 4, line 173: replace with levels Page 4, line 183 (revised document)-amended in the text
Page 9, line 291: delete this- Deleted in the text
Reviewer 2 Report
Javed et al. present RCT data on vitamin D effects on cardiovascular risk factors, hormones and liver markers in a small cohort of PCOS women. In general, the manuscript is well written and the topic is of interest .
Specific comments:
Abstract
-all abbreviations should be explained.
- The whole point with an RCT is to compare the placebo group and the intervention group. As stated in the title, the authors present RCT data. Therefore, only between-group comparisons should be presented in the abstract. Within group comparisons should be removed.
-line31: a p-value of 0.051 is not significant
-Line 37: “vitamin D deficient PCOS women”: Is vitamin D deficiency an inclusion criterion for this RCT?
Introduction:
line 60: ref. 11: menstrual frequency was improved in PCOS women receiving metformin or a combination of metformin+ vitamin D+calcium. Therefore it is not correct to state that vitamin D improved menstrual frequency as all women in group 1 and 2 also received metformin. There are recent RCTs comparing VD and placebo analysing menstrual frequency (Eur J Nutr. 2018 Jun 26. doi: 10.1007/s00394-018-1760-8)
line 61: ref12: This is a very small single arm open study including 12 women. The authors should focus on RCTs on vitamin D effects in PCOS women (summarized for example in Nutrients. 2018 Nov 2;10(11). pii: E1637. doi: 10.3390/nu10111637. or Endocr Connect. 2018 Mar;7(3):R95-R113)
line 76-77: vitamin D overweigh/deficient??
Material and Methods:
-What was the rationale for choosing a dose of 3200 IU daily?
-line 89-90: Why did the authors exclude PCOS women with T2DM but included women with metformin treatment? The authors cannot exclude that PCOS women taking metformin have T2DM.
-line 107: “including”
-Statistics: Did the authors adjust analyses for multiple testing (Bonferroni correction?)
-What was the primary outcome of the study? Why was the study powered to detect differences in CRP? Was CRP the primary outcome? Please clarify.
Discussion
-line187-189: The authors should clearly state that whether vitamin D effects were significant in within or between group comparisons. As the authors performed a RCT (as stated in the title), they should focus on between group comparisons.
-Line 269: As stated above, the authors cannot exclude that PCOS women taking metformin are affected by T2DM
-Line 271: “due to seasonal variation”: Did the authors analyse the impact of seasonality on their results?
-Did the authors perform statistical analyses after the exclusion of PCOS women taking metformin? Did the results remain stable in subgroup analyses in women without metformin?
General remarks:
-Did the authors assess calcium intake?
-The authors should focus on results from between-group comparisons.
-In the Introduction as well as in the Discussion section, the authors should focus on RCTs rather than on uncontrolled studies.
-information on study medication and placebo is missing (tablets, drops etc..)
-The authors should adhere to the CONSORT statement when reporting design results etc. (some items are missing: setting and location of recruitment, dat collection; patient flow chart; completely defined prespecified primary and secondary outcomes)
Author Response
Reviewer 2
Javed et al. present RCT data on vitamin D effects on cardiovascular risk factors, hormones and liver markers in a small cohort of PCOS women. In general, the manuscript is well written and the topic is of interest.
Thank you for your positive comments.
Specific comments:
Abstract
-all abbreviations should be explained. All abbreviations are now explained in the abstract
- The whole point with an RCT is to compare the placebo group and the intervention group. As stated in the title, the authors present RCT data. Therefore, only between-group comparisons should be presented in the abstract. Within group comparisons should be removed.
We agree with the reviewer and we have now removed the within group comparisons from the abstract. Please see revised abstract.
-line31: a p-value of 0.051 is not significant
We have now reported this as a weak effect in the abstract, but also throughout the manuscript (results and discussion).
-Line 37: “vitamin D deficient PCOS women”: Is vitamin D deficiency an inclusion criterion for this RCT?
Vitamin D deficiency was an inclusion criterion for this RCT. Please see inclusion criteria, page 2, lines 83-88 (revised manuscript): “Fifty-four reproductive-aged women (18–45 years) diagnosed with PCOS based on all three diagnostic criteria of the Rotterdam consensus were screened for vitamin D deficiency [26]. Non-classical 21-hydroxylase deficiency, hyperprolactinemia, Cushing’s disease and androgen-secreting tumours were excluded by appropriate tests. The participants were considered vitamin D-deficient, when their serum 25-hydroxyvitamin D (25OH-D) levels were less than 20 ng/mL (<50 nmol/L)”.
Introduction:
line 60: ref. 11: menstrual frequency was improved in PCOS women receiving metformin or a combination of metformin+ vitamin D+calcium. Therefore it is not correct to state that vitamin D improved menstrual frequency as all women in group 1 and 2 also received metformin. There are recent RCTs comparing VD and placebo analysing menstrual frequency (Eur J Nutr. 2018 Jun 26. doi: 10.1007/s00394-018-1760-8)
Thank you for your comment. We have misreported references in the initial version of the manuscript. We have now included the correct references by Irani et al., 2015 and Wehr et al. 2011, which support improvements in menstrual frequency. We have also added the recent RCT pointed out by the reviewer to support the conflicting findings of vitamin D supplementation studies.
Please see revision-page 2, lines 59-64:
“Studies on vitamin D supplementation in PCOS have yielded mixed results, with some of them suggesting beneficial effects on glucose metabolism and insulin resistance [10], especially when vitamin D is given continuously at lower (<4000 IU/day) doses as suggested in a recent meta-analysis [11], and improvements in menstrual frequency [12,13] and hyperandrogenism [13-15], whereas others demonstrate no significant improvements [16-20]”.
line 61: ref12: This is a very small single arm open study including 12 women. The authors should focus on RCTs on vitamin D effects in PCOS women (summarized for example in Nutrients. 2018 Nov 2;10(11). pii: E1637. doi: 10.3390/nu10111637. or Endocr Connect. 2018 Mar;7(3):R95-R113)
Thank you for your suggestions. We included the very recent meta-analysis by Lagowska et al (Nutrients. 2018 Nov 2;10(11). pii: E1637. doi: 10.3390/nu10111637and some RCTs summarised in this review. Please see relevant additions above and updated reference list.
line 76-77: vitamin D overweigh/deficient??
Please see amendment in the text, page 2, line 79-“overweight/obese vitamin D deficient women with PCOS.”
Material and Methods:
-What was the rationale for choosing a dose of 3200 IU daily?
Page 3, lines 107-110 (revised manuscript): The supplementation arm was based on the National Osteoporosis Society, UK guidelines which recommend treating patients who have serum 25OHD levels between 30–50 nmol/L and are at increased risk of developing vitamin D deficiency in the future because of reduced exposure to sunlight [30].
-line 89-90: Why did the authors exclude PCOS women with T2DM but included women with metformin treatment? The authors cannot exclude that PCOS women taking metformin have T2DM.
Metformin is a common prescription among women in PCOS, even in the absence of T2DM. All participants who were on metformin in the present study had Hba1c levels less than 42 mmol/mol. We have acknowledged that this is a limitation of the current study (page 8, line 278-282). Additionally, we have now rerun the between group comparisons after excluding the participants on metformin in both groups and our results did not change, suggesting that metformin did not interfere with our intervention in the present study.
We have added relevant descriptions in our results and discussion:
Page 5, lines 191-194 (revised manuscript): “Given that some participants in the vitamin D (n=3) and the placebo group (n=8) were on metformin, which may interfere with our results, we repeated between group comparisons after excluding data from participants taking metformin and the results remained unchanged (data not shown)”.
Page 8, lines 294-296 (revised manuscript): “The results of the between group comparison remained unchanged after excluding the participants taking metformin in both groups, further supporting the notion that metformin did not interfere with our results”.
-line 107: “including” Page 3, line 110 (revised document)-amended in the text
-Statistics: Did the authors adjust analyses for multiple testing (Bonferroni correction?)
One of the statistical issues to address is the problem of multiple testing when many variables are present and the possible inflation of type I error, however, there is no consensus on what procedure to adopt to allow for multiple comparisons. Hence, we followed the recommendation of Rothman 1990 and have not adjusted for this (reference: Rothman KJ. No adjustments are needed for multiple testing. Epidemiology 1990; 1:43-46).
-What was the primary outcome of the study? Why was the study powered to detect differences in CRP? Was CRP the primary outcome? Please clarify.
We confirm that the study was powered to detect differences in hs-CRP, which was one of the primary outcomes in the study. This decision was based upon previous data produced in our research group (please see statistical analysis section).
Discussion
-line187-189: The authors should clearly state that whether vitamin D effects were significant in within or between group comparisons. As the authors performed a RCT (as stated in the title), they should focus on between group comparisons.
We have clearly stated which results refer to within and between group comparisons. Given that our study is a RCT, we have emphasised upon and reported first the results of the between group comparisons.
Page 7, lines 202-207 (revised manuscript): “In this study, we demonstrate that compared to placebo, vitamin D supplementation resulted in modest improvements in ALT and insulin resistance, whereas no further between-group differences were seen in cardiovascular risk factors or hormones. Within group comparison also showed that supplementation with vitamin D (3200 IU/day) over a 3-month period resulted in significant improvements in individual liver markers (HA, ALT) and in the ELF score compared to baseline in overweight/obese, vitamin D deficient women with PCOS.”
-Line 269: As stated above, the authors cannot exclude that PCOS women taking metformin are affected by T2DM
Please see our response on this question above.
-Line 271: “due to seasonal variation”: Did the authors analyse the impact of seasonality on their results?
Given that the study started in July 2015 and the last patient, last visit was in November 2016, we speculate greater, overall exposure to sunlight during the study duration compared to winter months. From a statistical perspective, the analysis of seasonality is a very specialized area of statistics (time-series) in which we have no expertise.
-Did the authors perform statistical analyses after the exclusion of PCOS women taking metformin? Did the results remain stable in subgroup analyses in women without metformin?
Based on the comments provided by both reviewers, we reran the between group comparisons after excluding the participants on metformin in both groups and our results did not change, suggesting that metformin did not interfere with our intervention in the present study.
We have added a relevant point in our results and discussion:
Page 5, lines 191-194 (revised manuscript): “Given that some participants in the vitamin D (n=3) and the placebo group (n=8) were on metformin, which may interfere with our results, we repeated between group comparisons after excluding data from participants taking metformin and the results remained unchanged (data not shown)”.
Page 8, lines 294-296 (revised manuscript): “The results of the between group comparison remained unchanged after excluding the participants taking metformin in both groups, further supporting the notion that metformin did not interfere with our results”.
General remarks:
-Did the authors assess calcium intake?
Unfortunately, we did not assess calcium intake as part of this study. This is an aspect we will consider in our future work in this population.
-The authors should focus on results from between-group comparisons.
As per reviewer’s suggestion, we have now clarified between/within group comparisons and we have emphasised upon between group comparisons. Please see our responses to specific comments above.
-In the Introduction as well as in the Discussion section, the authors should focus on RCTs rather than on uncontrolled studies.
Based on reviewer’s recommendation, we have added relevant RCTs and meta-analyses. Please see our response on earlier comments regarding the introduction and discussion, as well as the updated reference list.
-information on study medication and placebo is missing (tablets, drops etc..)
We have now added this information, please see page 2, lines 102-104 “Forty eligible patients were randomly assigned to the vitamin D 3200 IU (Swiss Caps AG, Cornu, Switzerland) or placebo (Swiss Caps AG, Kirchberg, Switzerland) in capsule form taken once daily”.
-The authors should adhere to the CONSORT statement when reporting design results etc. (some items are missing: setting and location of recruitment, data collection; patient flow chart; completely defined prespecified primary and secondary outcomes)
Thank you for this comment, details on these items can be found below:
Setting and location of recruitment and data collection: Page 2, lines 83-84 (revised manuscript)- “This double-blind, randomized, placebo-controlled study was performed in the Academic Diabetes, Endocrinology and Metabolism Unit, Hull University Teaching Hospitals NHS Trust, Hull, United Kingdom”.
Patient flow chart: Figure 1
Primary and secondary outcomes: Page 3, lines 110-116 (revised manuscript)- “High-sensitivity C-reactive protein (hs-CRP), lipid profile, glucose levels, insulin levels and the homeostatic model assessment-insulin resistance (HOMA-IR) were assessed as the primary outcome measures. As secondary outcomes, other cardiovascular risk factors [weight, body mass index (BMI),], hormones [free androgen index (FAI), testosterone, sex hormone binding globulin (SHBG)] and liver markers [alanine aminotransferase (ALT), hyaluronic acid (HA), N-terminal propeptide of type III procollagen (PIIINP), tissue inhibitor of metallo-proteinases-1 (TIMP-1) and the enhanced liver fibrosis (ELF) score] were evaluated”.

Round 2
Reviewer 2 Report
Many improvements have been made. The authors have answered all the open questions.